# CNN$^2$: Viewpoint Generalization via a Binocular Vision

**Wei-Da Chen**
Department of Computer Science
National Tsing-Hua University
Taiwan, R.O.C.
`wdchen@datalab.cs.nthu.edu.tw`

**Shan-Hung Wu**
Department of Computer Science
National Tsing-Hua University
Taiwan, R.O.C.
`shwu@cs.nthu.edu.tw`

## Abstract

The Convolutional Neural Networks (CNNs) have laid the foundation for many techniques in various applications. Despite achieving remarkable performance in some tasks, the 3D viewpoint generalizability of CNNs is still far behind humans visual capabilities. Although recent efforts, such as the Capsule Networks, have been made to address this issue, these new models are either hard to train and/or incompatible with existing CNN-based techniques specialized for different applications. Observing that humans use binocular vision to understand the world, we study in this paper whether the 3D viewpoint generalizability of CNNs can be achieved via a binocular vision. We propose CNN$^2$, a CNN that takes two images as input, which resembles the process of an object being viewed from the left eye and the right eye. CNN$^2$ uses novel augmentation, pooling, and convolutional layers to learn a sense of three-dimensionality in a recursive manner. Empirical evaluation shows that CNN$^2$ has improved viewpoint generalizability compared to vanilla CNNs. Furthermore, CNN$^2$ is easy to implement and train, and is compatible with existing CNN-based specialized techniques for different applications.

## 1 Introduction

Convolutional Neural Networks (CNNs, LeCun et al. (1989, 1998)) are models inspired by how the animal visual cortex works (Hubel and Wiesel (1962)) and are computationally modelled (Fukushima and Miyake (1982)) based on local connectivities between neurons and hierarchically organized transformations of an image. CNNs have greatly advanced the state-of-the-art performance of visual recognition tasks, such as image classification (Real et al. (2018); He et al. (2016); Krizhevsky et al. (2012)), localization and detection (Lin et al. (2017b); Redmon et al. (2016)), segmentation (He et al. (2017); Long et al. (2015)), and have driven the development of various specialized techniques for applications in natural language processing (Gehring et al. (2017a,b)), search (McDonald et al. (2018); Dai et al. (2018)), mapping (Liu et al. (2017); Zhu et al. (2017)), medicine (Esteva et al. (2019)), drones (Kim et al. (2017); Kyrkou et al. (2018)), and self-driving cars (Codevilla et al. (2018); Bojarski et al. (2016)).

Despite giving impressive performance in many applications, CNNs still have a long way to go in terms of being comparable to human's visual ability. One important aspects where vanilla CNNs fall short is referred to as *transformation generalizability*—the ability to generalize what have been learned from training images to understand the transformed images at test time. While there are many studies (Jaderberg et al. (2015); Maninis et al. (2016); Cheng et al. (2016); Laptev et al. (2016); Worrall et al. (2017); Hinton et al. (2018); Cheng et al. (2019); Ecker et al. (2019)) that address 2D transformations (e.g., rotation, scaling, and sheering), few efforts have been made towards a more challenging goal called *3D viewpoint generalization*; that is, to understand images of 3D objects with *unseen* viewpoint translation at test time.

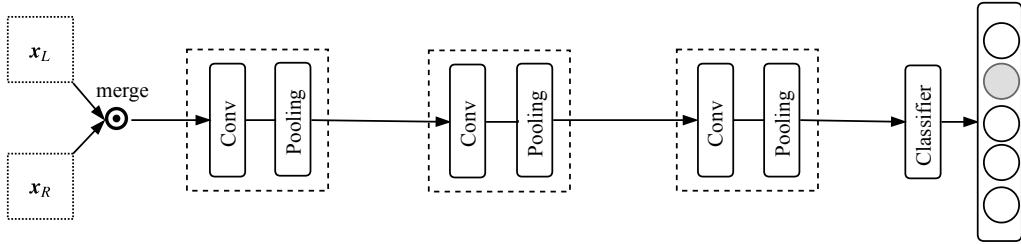

Figure 1: A simple extension (LeCun et al. (2004)) of the CNN architecture for binocular images. The two images from the left eye and the right eye are merged and then fed into regular CNN layers.

A well-known branch of studies targeting 3D viewpoint generalizability is the capsule networks (Hinton et al. (2011); Sabour et al. (2017); Hinton et al. (2018)), which represent an object or a part of an object as a collection of neurons called a capsule. It organizes different capsules in a parse tree where the output of lower-level capsules is dynamically routed to upper-level capsules using an agreement protocol. The capsule networks show some promising results that deserve further investigation. Some researchers who have done so, such as Peer et al. (2018), found out that capsule networks are harder to train than conventional CNNs, because the capsules increase the number of model parameters. Also, the iterative routing-by-agreement algorithm used for training is time consuming and does not ensure the emergence of a parse tree in the network. Additionally, the architecture of capsule networks is not compatible with CNNs, which prevents the large CNN ecosystem from being able to add values to and benefit from the capsule nets.

The above drawbacks motivate us to seek for a more generalizable model that is compatible with existing CNN-based techniques. An obvious difference between how humans and machines view an object is that humans visualize using two eyes. Fortunately nowadays, binocular images can be easily collected. For instance, majority of people are using their smartphones, which are now usually equipped with dual or more lens (Moura et al. (2014)), as cameras to record daily events. As another example, one can extract two nearby frames in online videos to construct a large binocular image dataset.

In this paper, we propose CNN$^2$, a convolutional neural network with improved 3D viewpoint generalizability by taking two binocular images as input. Unlike a simple CNN extension (LeCun et al. (2004), as shown in Figure 1) that stacks up two images along the channel dimension and then feeds them to a regular CNN network, CNN$^2$ explicitly models some priors from binocular vision. We apply contrastive channel augmentation to the respective images so they are scanned by filters (or kernels) in *two* parallel, complementary feedforward pathways. This resembles the dual-path central visual pathways (Wurtz et al. (2000); Milner and Goodale (2006)) in human brains. After the augmentation, the CNN$^2$ employs a novel *concentric multi-scale pooling* layers that are applied *before* the convolutional layers to learn the in-focus and out-of-focus features. Such a design is inspired by the interactions between the V1 and V2 visual cortices in human visual cortex system (Biederman (1987); Reid and Alonso (1995); Murphy et al. (1999)). We conduct experiments using binocular images from the SmallNORB (LeCun et al. (2004)), ModelNet (Wu et al. (2015)) and larger-scale RGB-D Object (Lai et al. (2011)) datasets. The results demonstrate that CNN$^2$ can learn a sense of three-dimensionality in a recursive manner and has improved 3D viewpoint generalizability. Furthermore, CNN$^2$ is easy to implement and train, and is compatible with existing CNN-based specialized techniques for different vision applications.

## 2   Model Design of CNN$^2$

For ease of presentation, we consider a supervised learning task: given a task model $f$ and a binocular image set $\mathcal{D} = (\mathcal{X}, \mathcal{Y}) = \{(\boldsymbol{x}_{\mathrm{L}}^{(i)}, \boldsymbol{x}_{\mathrm{R}}^{(i)}, \boldsymbol{y}^{(i)})\}_i$ where each $\boldsymbol{x}_{\mathrm{L}}^{(i)}$ and $\boldsymbol{x}_{\mathrm{R}}^{(i)}$ represent the images taken from the left eye and the right eye viewpoints, respectively. Our goal is to design an embedding model $g$ such that, after being trained using $\mathcal{D}$, it can help $\hat{\boldsymbol{y}}' = f(g(\boldsymbol{x}_{\mathrm{L}}', \boldsymbol{x}_{\mathrm{R}}'))$ predict the correct label $\boldsymbol{y}'$ of a pair $(\boldsymbol{x}_{\mathrm{L}}', \boldsymbol{x}_{\mathrm{R}}')$ of binocular images taken from an *unseen* viewpoint at test time.

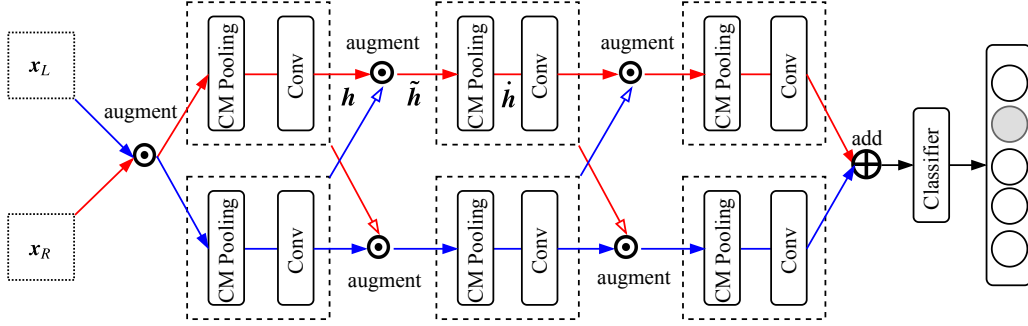

Figure 2: CNN$^2$ model architecture that has two feedforward pathways providing the dual parallax augmentation at different abstraction levels. $\boldsymbol{h} \in \mathbb{R}^{W \times H \times C}$: raw feature map. $\tilde{\boldsymbol{h}} \in \mathbb{R}^{W \times H \times 2C}$: parallax augmented feature map. $\dot{\boldsymbol{h}} \in \mathbb{R}^{W \times H \times 2CS}$: output of the concentric multi-scale (CM) pooling, where $S$ is the number of scales.

One naive idea to improve the 3D viewpoint generalizability is to learn a depth map (Godard et al. (2017); Kendall et al. (2017)) from a pair of binocular images, treat the depth map as a new channel in the input (left or right eye) image, and feed the augmented image to a regular CNN just like the one shown in Figure 1. However, the depth information is only a subset of the knowledge that can be learned from binocular vision. Studies in neuroscience have found out that human's visual system can detect stereoscopic edges (Von Der Heydt et al. (2000)), foreground and background (Qiu and Von Der Heydt (2005); Maruko et al. (2008)), and illusory contours of objects extrapolated from seen angles (von der Heydt et al. (1984); Anzai et al. (2007)) from binocular images. Hence, our goal is to design a model $g$ that is able to capture these generic patterns.

Next, we present the CNN$^2$ that is able to recognize generic binocular vision patterns recursively at different layers. It can be jointly trained with the task model $f$ in an end-to-end manner.

**Dual Feedforward Pathways.** Figure 2 shows the architecture of CNN$^2$. Unlike a regular CNN that has only one feedforward pathway, the CNN$^2$ employs two parallel, yet complementary, feedforward pathways for the left and right eye images, respectively. At each layer, the binocular images or feature maps are combined and then split by following the *dual parallax augmentation* procedure. Specifically, given a pair of binocular images or feature maps ($\boldsymbol{h}_{\mathrm{L}} \in \mathbb{R}^{W \times H \times C}, \boldsymbol{h}_{\mathrm{R}} \in \mathbb{R}^{W \times H \times C}$), we augment $\boldsymbol{h}_{\mathrm{L}}$ by adding the parallax $\boldsymbol{h}_{\mathrm{R}} - \boldsymbol{h}_{\mathrm{L}}$ as new channels. Similarly, we augment $\boldsymbol{h}_{\mathrm{R}}$ by $\boldsymbol{h}_{\mathrm{L}} - \boldsymbol{h}_{\mathrm{R}}$. The two augmented maps ($\tilde{\boldsymbol{h}}_{\mathrm{L}} \in \mathbb{R}^{W \times H \times 2C}, \tilde{\boldsymbol{h}}_{\mathrm{R}} \in \mathbb{R}^{W \times H \times 2C}$) contain the information from both eyes, but on different bases (defined by the three original channels). Then each augmented map is fed into the next layer either through the left or right pathway. This allows the filters (or kernels) in convolutional layers to recursively detect stereoscopic features at different abstraction levels by looking into the parallax. The small differences between the two input images at the pixel level and at shallow layers may add up to a big difference at a deeper layer, as discovered in human visual system (Biederman (1987); Murphy et al. (1999); Reid and Alonso (1995)).

**Concentric Multi-Scale Pooling**: Human and camera lens both reflect the light following the principles of optics, and objects become blurry when they are out of focus. In addition to parallax augmentation, by comparing clear and blurred features from the previous layer, we allow a filter to detect stereoscopic patterns. We introduce a new type of pooling layers, called the *concentric multi-scale* (CM) *pooling*. Figure 3 shows how the CM pooling works. Formally, let $\tilde{\boldsymbol{h}} \in \mathbb{R}^{W \times H \times 2C}$ be an augmented image or feature map and suppose there are $S$ given scales. At each scale $s = 0, 1, \cdots, S-1$, we first obtain a temporary map $\boldsymbol{e}^{(s)} \in \mathbb{R}^{W \times H \times 2C}$ (assuming zero padding), where

$$e_{i,j,c}^{(s)} = \mathrm{pool}_{p,q:i-s \leq p \leq i+s \text{ and } j-s \leq q \leq j+s}\{\tilde{h}_{p,q,c}\}$$

and pool$\{\cdot\}$ is a pooling operation (e.g., $\max\{\cdot\}$ or avg$\{\cdot\}$). Then, these temporary maps are stacked up along the channel dimension to produce $\dot{\boldsymbol{h}} \in \mathbb{R}^{W \times H \times 2CS}$. Unlike conventional pooling layers that come after the convolutional layers, the CM pooling layers are placed *before* the convolutional layers. This aids the filter in the next layer to easily detect stereoscopic patterns, by contrasting blurry features with clear features. The translation invariance created by an $\boldsymbol{e}^{(s)}$ at a large scale ($s$)

detects blurry features in the background, while an $e^{(s)}$ at a small scale detects clear features in the foreground.

Note that the feature map $\dot{h}$ produced by a CM pooling layer is equivariant to input translation. The CNN² does *not* use conventional pooling layers that are known to introduce translation invariance and decrease viewpoint generalizability (Hinton et al. (2011); Sabour et al. (2017)). A drawback of the CNN² is that a feature map at a hidden layer will have the same (large) width and height as that of the input image, which could slow down the speed of computation. Additionally, there is an increase in the number of filter weights due to a larger number of channels ($2CS$) in $\dot{h}$. These problems can be mitigated by using fewer filters at each layer. Empirically, we found that the CNN² requires much fewer filters than the conventional CNNs for the same satisfactory performance. Also note that CNN² does not modify the convolutional layers in regular CNNs. This means that the CNN² is compatible with the existing convolution-based enhancement techniques and can contribute to and benefit from the rich CNN ecosystem.

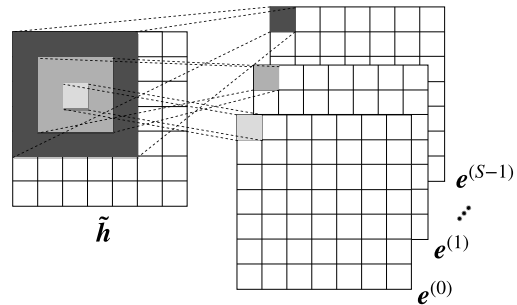

Figure 3: Concentric multi-scale pooling that is placed *before* a convolution in the CNN². It enables a filter to easily detect stereoscopic patterns by contrasting in-focus features with out-of-focus features.

**Inspiration from Human Visual System.** While the effectiveness of the CNN² solely depends on engineering efforts, the design of CNN² model is loosely inspired by how the human visual system works. Figure 4 shows an oversimplified version of the mammals' visual system (Wurtz et al. (2000); Milner and Goodale (2006)). The visual information mainly flows through the *central visual pathways* in the brain. Although recent studies (Kheradpisheh et al. (2016); Wallis et al. (2017); Laskar et al. (2018); Long and Konkle (2018)) have found correspondence between the activations of CNN layers and the neuron responses in human's visual cortex system, the CNNs are still different from human visual system in many ways. One key difference is that the CNNs have only one feedforward pathway. On the other hand, the CNN² employs two feedforward pathways, which resembles the left and right halves of the central visual pathway in two sides of our brains. The dual parallax augmentation at the input layer of the CNN² corresponds to the optic chiasma in human visual system, where the information coming from both eyes is combined, augmented, and then split. At deep layers, it resembles the interactions between the left and right sides of the brain which are known to have their own bias (Gotts et al. (2013)). For more discussion about the correspondency between the CNN² components and human visual system, please refer to Section 2 of the supplementary materials.

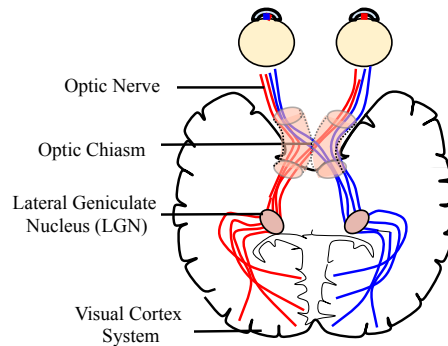

Figure 4: Visual system of mammals (Wurtz et al. (2000); Milner and Goodale (2006)). The electrical pulses from the two eyes are merged at the optic chiasma and then sent to the right and left brains separately following two visual pathways. The pulses will then finally reach the visual cortex system (Biederman (1987); Reid and Alonso (1995); Murphy et al. (1999); Gotts et al. (2013)) where the visual image is heavily processed by the interaction between the right and left brains with respective bias.

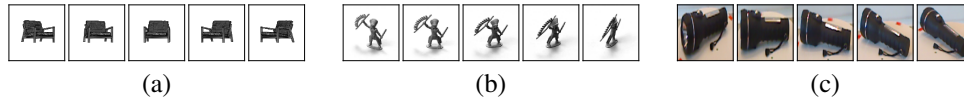

(a)                          (b)                          (c)

Figure 5: Examples of left eye images taken from different viewpoints in (a) ModelNet2D dataset (chairs), (b) SmallNORB dataset (humans), and (c) RGB-D Object dataset (flashlights).

# 3   Further Related Work

Here, we review further related works that are not mentioned in Sections 1 and 2. For a complete discussion of the related work, please refer to Section 1 of the supplementary materials.

**3D Viewpoint Generalization.** In addition to the capsule networks (Hinton et al. (2011); Sabour et al. (2017); Hinton et al. (2018)), another way to viewpoint generalization is using voxel discretization (Su et al. (2015); Qi et al. (2016); Yan et al. (2016); Qi et al. (2017)), which reconstructs manifold (and non-manifold) surfaces in the 3D space from point clouds using voxels as an intermediate representation . However, these models require either the voxel-level supervision or omnidirectional images as input, which are both expensive to collect in practice. **Binocular Vision.** Binocular images have been used for learning the depth information. Godard et al. (2017) utilize binocular images to make a model learn the depth map in an unsupervised manner. Kendall et al. (2017) exploit the geometry and context information in binocular images to let a model learn the disparity map of a stereogram. However, few studies (LeCun et al. (2004), whose architecture is shown in Figure 1) have been made to understand the impact of binocular vision on CNN generalizability. **Multi-Scale Feature Representations.** A $CNN^2$ layer extracts features at multiple scales, thus is related to the work on multi-scale feature learning (Yang and Ramanan (2015); Cai et al. (2016); Lin et al. (2017a); Chen et al. (2019)). Unlike most existing models that concatenate the multi-scale features to learn patterns, $CNN^2$ pools (via the CM pooling) multi-scale features to make them of equal size and then stack them up along the channel dimension. The location information encoded in different feature maps are aligned. This allows the next convolutional layer to learn location independent patterns (and a sense of 3D dimensionality) by contracting the features at different scales. **Pooling Strategies.** Our CM pooling is cosmetically similar to some existing pooling techniques (He et al. (2014); Gong et al. (2014); Qi et al. (2018)). The spatial pyramid pooling (He et al. (2014)) pools image pixels using predefined patches, which require domain-specific knowledge to define. The multi-scale orderless pooling (Gong et al. (2014)) outputs feature maps of different sizes, but these maps are not "zoomed" to equal size and then stacked up along the channel dimension to help the filters contract features at different scales at the same location. Qi et al. (2018) propose a concentric circle pooling strategy to achieve rotation invariance, where multiple filters scan an image or feature map following concentric window-sliding paths. Their term "concentric" is different from ours.

# 4   Experiments

In this section, we evaluate the performance of $CNN^2$ using three binocular image datasets: 1) the ModelNet2D dataset rendered from ModelNet40 (Wu et al. (2015)) following the settings used by LeCun et al. (2004), 2) the SmallNORB dataset (LeCun et al. (2004)), and 3) the RGB-D Object dataset (Lai et al. (2011)), which consist of 12,311 grayscale, 48,600 grayscale, and 250,000 color images taken from different azimuths with 5-, 20-, and 10-degree ticks, respectively. Figure 5 shows some example images from these datasets. Only the SmallNORB dataset provides binocular images. For the rest of the datasets, we use pairs of images having successive azimuths degrees to simulate binocular images. We also sample 5 classes of objects from each dataset that look different from each other in any azimuths degree. For more information about the datasets and preprocessing, please refer to Section 3.1 of the supplementary materials. Note that 3D viewpoint generalization is a difficult and challenging problem, wherein majority of existing work were only evaluated on grayscale datasets. To the best of our knowledge, this is the first work that conducted experiments on colored datasets for 3D viewpoint generalization.

We implement $CNN^2$ and the following baselines using TensorFlow (Abadi et al. (2016)). **Vanilla CNN.** This is a simple CNN extension (LeCun et al. (2004)) whose architecture is shown in Figure 1. **CapsuleNet.** This is capsule network with EM routing (Hinton et al. (2018)). It uses the matrix capsules to capture the activation along with a pose matrix.   **PTN.** The perspective transformer

Table 1: The number of parameters in different models for the grayscale (ModelNet2D and Small-NORB) and RGB-D Object datasets.

| | Vanilla CNN | BL-Net | Monodepth | PTN | CapsuleNet | $CNN^2$ | $CNN^2$+BL |
|---|---|---|---|---|---|---|---|
| Grayscale | 333K | 411K | 19M+333K | 12M | 362K | 341K | 407K |
| RGB-D Object | 421K | 489K | 19M+427K | 12M | 568K | 493K | 506K |

Table 2: Average test accuracy of different models over *unseen* viewpoints and the time required to train these models. The pair of numbers in Monodepth denotes the training time for the depth map generator (stage 1) and CNN (stage 2), respectively. The training of PTN and CapsuleNet on the RGB-D dataset did not converge.

| | ModelNet2D | | SmallNORB | | RGB-D Object | |
|---|---|---|---|---|---|---|
| Models | Acc. (Unseen) | Time (min) | Acc. (Unseen) | Time (min) | Acc. (Unseen) | Time (min) |
| Vanilla CNN | 0.907 | 138 | 0.722 | 231 | 0.795 | 313 |
| BL-Net | 0.903 | 109 | 0.751 | 192 | 0.829 | 288 |
| Monodepth | 0.910 | 143+127 | 0.783 | 168+150 | 0.802 | 612+301 |
| PTN | 0.879 | 159 | 0.714 | 273 | 0.427 | - |
| CapsuleNet | 0.921 | 478 | 0.835 | 1328 | 0.476 | - |
| $CNN^2$ | **0.941** | **91** | **0.865** | **121** | **0.868** | **236** |
| $CNN^2$+BL | 0.918 | 115 | 0.787 | 251 | 0.778 | 315 |

network (Yan et al. (2016)) that outputs 3D voxels. The original paper assumes omnidirectional images of an object as the input. Here, we feed only the images within a particular range of view angles that is available at training time (see Section 4.1 for more details about the range) to the network to get output voxels. Then, we feed the voxels into a 3D convolutional neural network for classification. We follow the settings described in the original paper (Yan et al. (2016)) and the study (Maturana and Scherer (2015)) to train the entire model from end to end. **Monodepth.** A model based on the depth information, which is explicitly learned from the binocular images. The original Monodepth network (Godard et al. (2017)) is a model that outputs the depth map for a given pair of binocular images. It can be trained in an unsupervised manner. We create a two-stage training process here. In the first stage (pre-training stage), we train a Monodepth network and use it to generate a depth map. Then, in the second stage, we add the depth map into the the left eye image as an additional channel and feed the augmented image to a CNN. The CNN architecture is the same as that used in Vanilla CNN. We follow the settings described in the Monodepth network paper (Godard et al. (2017)) to train the model for stage one. **BL-Net.** This network is composed of a concatenation of Big-Little module (BL-module), which aims to extract multi-scale feature representations with a good trade-off between speed and accuracy. Here, we extend the Vanilla CNN by replacing its architecture with the Big-Little network following the settings in Chen et al. (2019). **$CNN^2$+BL.** To see whether our CM pooling can help a model learn beyond the multi-scale features, we also replace the blocks of layers of $CNN^2$ with the BL-modules, while keeping the dual feedforward pathways and parallax augmentation.

We conduct experiments on a computer with an Intel Core i7-6900K CPU, 64 GB RAM, and an NVIDIA Geforce GTX 1070 GPU. We did *not* augment the data at training time in order to observe the unbiased generalizability of different models. For each of the above models, we search for the best architecture for a given dataset. Table 1 shows the number of parameters in different models. Please see Section 3.2 of the supplementary materials for more details.

## 4.1 3D Viewpoint Generalization

To test the 3D viewpoint generalizability of different models, we train the models using (binocular) images taken from a limited range of view angles and then test the model performance using images taken from unlimited view angles. On the ModelNet2D dataset, we use the images taken from azimuths of degrees from 50 to 125 as the training set, degrees from 30 to 45 and from 130 to 145 as the validation set, and unlimited degrees as the test set. On the SmallNORB dataset, we use the images taken from azimuths of degrees from 20 to 80 as the training set, degrees at 0 and 100 as the validation set, and the rest as the test set. On the RGB-D Object dataset, images of different objects are taken from different viewpoints. So, we use images taken from one third of continuous

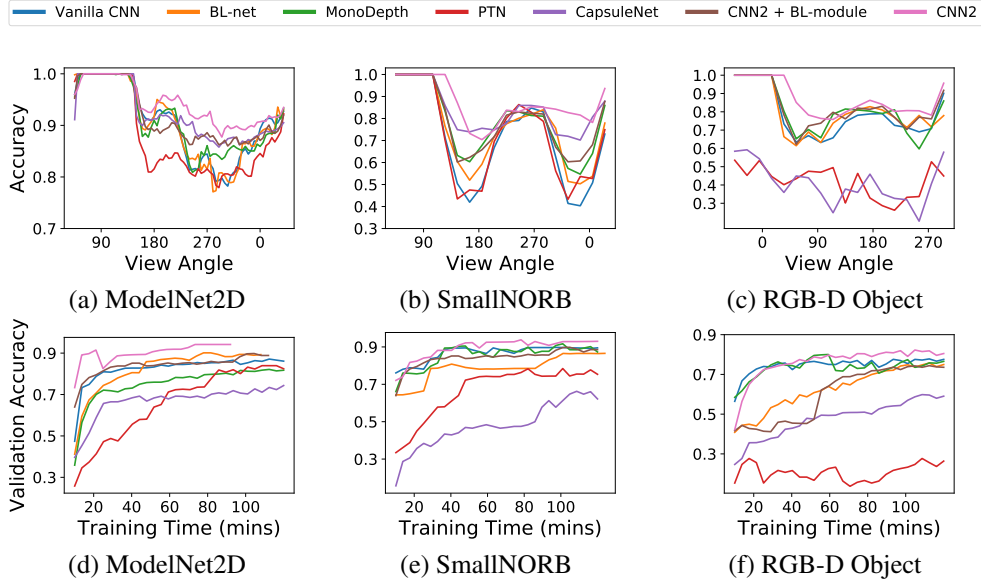

Figure 6: 3D viewpoint generalizability of models trained on each dataset. (a)-(c) Test accuracy at different view angles with about two third of the view angles that are *not* seen at training time. (d)-(f) Learning curve (early stopped or truncated at 120 mins).

viewpoints of each object as the training set and the remaining images as the test set. We further split one third of the training images having continuous viewpoints as the validation set.

The average test accuracy of different models over all unseen angles and the time required to train these models is shown in Table 2. It can be seen that $CNN^2$ achieves higher accuracy than all the baseline models. It also converges faster during the training process. Figure 6 shows how the accuracy of different models varies at different view angles and how the models learn over time. On grayscale datasets (ModelNet2D and SmallNORB), both the CapsuleNet and $CNN^2$ give significantly better performance than the other baselines at challenging view angles where the objects look very different from what they appeared at training time. However, the $CNN^2$ is much faster to train than the CapsuleNet. In fact, the learning speed of the $CNN^2$ is even faster than the Vanilla CNN. Note that the $CNN^2$ uses much fewer filters (50) than the Vanilla CNN (112). This justifies that the patterns detected by $CNN^2$ filters are useful for 3D viewpoint generalization. Also, by comparing the performance of $CNN^2$ and $CNN^2$+BL, we know that the performance gain is not from merely extracting the multi-scale features. The CM pooling indeed helps the $CNN^2$ filters learn generic stereoscopic features by contracting the features at different scales. On the colored RGB-D Object dataset, the $CNN^2$ still outperforms other baselines. The CapsuleNet and PTN perform poorly in this case. We have searched different architectures for these models for better performance, but failed (see Section 3.2 of the supplementary materials). Our findings about the CapsuleNet is consistent with Peer et al. (2018), who pointed out that the capsule networks are harder to train and the iterative routing-by-agreement algorithm used for training does not ensure the emergence of a parse tree in the networks. As for the (voxel-based) PTN, we suspect that it has too high sample complexity to easily learn from a color binocular dataset.

### 4.2 Backward Compatibility

**2D Rotation Generalizability.** The $CNN^2$ does not change the convolution operation, which makes it compatible with the rich CNN ecosystem. To see how this can be beneficial, we design a more challenging task where the models are asked to predict the labels of images taken from unseen view angles *and unseen 2D rotations* at test time. We train and validate the models using the images from the ModelNet2D dataset that have 50% chance to be rotated 90 degrees clockwisely. At test time, we feed the model with the images that is rotated either 180 or 270 degrees clockwisely (in addition to viewpoint shift described in Section 4.1). Without data augmentation and specialized techniques, the convolution-based methods, including $CNN^2$, give degraded performance in this

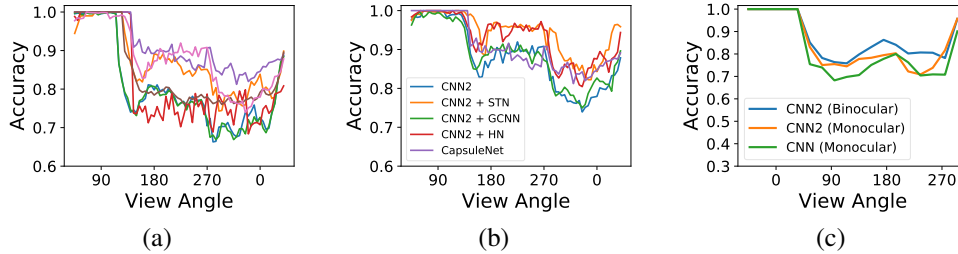

Figure 7: Backward compatibility. (a) 3D viewpoint + 2D rotation generalizability of models trained on the ModelNet2D dataset. Test accuracy for objects with *unseen* rotations (180 and 270 degrees) at different view angles. Angles outside $[50, 125]$ are unseen. See Figure 6 for the legends. (b) $CNN^2$ is backward compatible with existing CNN-based techniques and can be readily enhanced to have 2D rotation generalizability. (c) Performance of $CNN^2$ with monocular images from the RGB-D Object dataset. $CNN^2$ is also backward compatible with single-eye image classification tasks.

task, as shown in Figure 7(a). Only the non-convolutional CapsuleNet achieves stable performance across viewpoints and rotations. However, there exists many CNN-based techniques that target 2D rotation generalizability, such as the spatial transformer networks (STN, Jaderberg et al. (2015)), group equivariant convolutional neural networks (GCNN, Cohen and Welling (2016)), and harmonic networks (HN, Worrall et al. (2017)). We integrate these methods into the $CNN^2$ and get significantly better performance, as shown in Figure 7(b). The performance boost is consistent on other datasets (see Section 3.3 of the supplementary materials). This demonstrates the potential of $CNN^2$ for benefitting from, and contributing to, many applications where CNNs thrive. **Monocular Images.** With monocular images, the parallax channels contain all zeros, therefore the $CNN^2$ degenerates into a conventional CNN gracefully. Figure 7(c) shows the performance of degenerated $CNN^2$ with the single-eye images from the RGB-D Object dataset. Although the degenerated $CNN^2$ with monocular images does not outperform the fully functional $CNN^2$ with binocular inputs due to the lack of binocular information, its performance is comparable with (if not surpasses) that of vanilla CNN because it models more prior than vanilla CNN. The $CNN^2$ is compatible with single-eye image classification tasks.

## 4.3 More Experiments

**Ablation Study.** Here, we investigate whether each designed component used by the $CNN^2$ improves 3D viewpoint generalizability. Following the settings described in Section 4.1, we compare the $CNN^2$ with its variant where the weights along the dual feedforward pathways are tied. The results, as shown in Figure 8(a), indicate that having two feedforward pathways is indeed beneficial. Next, we compare the $CNN^2$ with another version where parallax augmentation is dropped. As we can see from Figure 8(b), the parallax augmentation can improve the model generalizability at challenging view angles. Next, we test whether the concentric multi-scale (CM) pooling contributes to 3D viewpoint generalizability. We compare the $CNN^2$ with a variant where the CM pooling layers are replaced by conventional max pooling layers. The results, which are shown in Figure 8(c), confirm its effectiveness. We can also see from Figure 8(d) that the standalone CM pooling is sufficient to improve the generalizability of vanilla CNN. **Pooling before Convolution.** We also have an interesting observation: while placing the pooling layers after the convolution layers give better performance in regular CNNs, it hurts the generalizability of $CNN^2$, as shown in Figure 8(e). This reminds us that something we took for granted in monocular vision may not be the best choice for the binocular cases. **Fusion of the Two Feedforward Pathways.** To show that the fusion (i.e., dual parallax augmentation) of the two feedforward pathways at each layer is beneficial, we compare $CNN^2$ with two new baselines that perform early and late fusion in only the first and last layer, respectively. Figure 8(f) shows the results on the RGB-D Object dataset. $CNN^2$ outperforms other baselines because it has fusion at all layers, which allows small differences between the feature maps in two paths to add up to a big difference at a deeper layer. **Backbone Choices.** The $CNN^2$ can work with different backbone architectures. To show this, we compare the performance of $CNN^2$ with ResNet-50 and a toy ResNet as the backbone on the SmallNORB dataset. The toy ResNet is consisted of 2 residual blocks and has similar number of parameters as $CNN^2$. The results are shown in Figure 9(a). Although the SmallNORB dataset contains only grayscale images and looks easy,

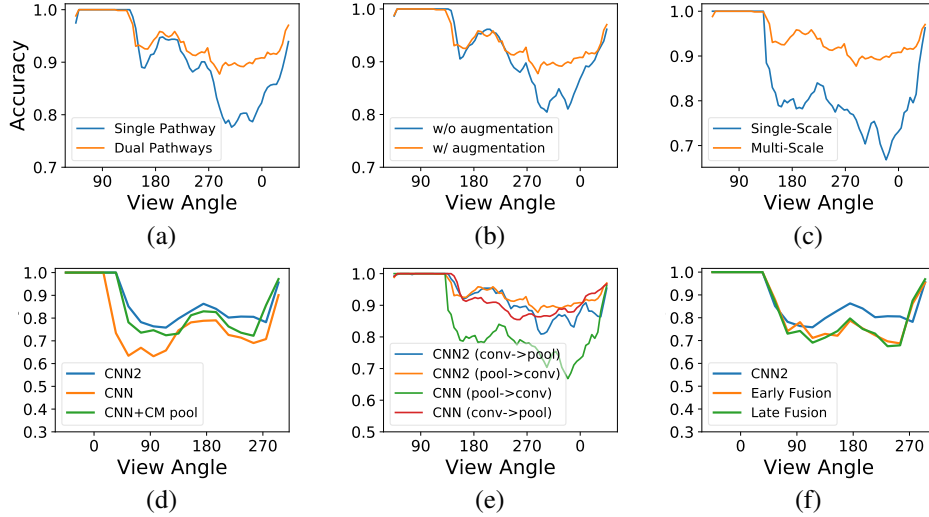

Figure 8: Ablation Study. (a) Single (weight-tied) vs. dual feedforward pathways. (b) CNN$^2$ with vs. without parallax augmentation. (c) Max pooling (before convolution) vs. CM pooling. (d) The CM pooling, by itself, can improve the performance of vanilla CNN on the RGB-D Object dataset. (e) Performing pooling before convolution improves performance in CNN$^2$, but not in vanilla CNN. (f) Performance of CNN$^2$ variants with different fusion strategies on the RGB-D Object dataset.

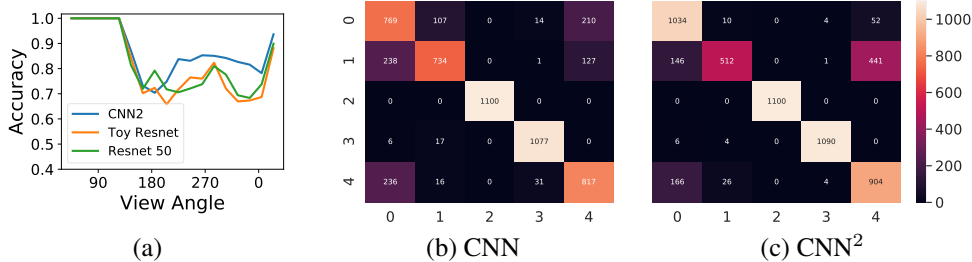

Figure 9: (a) Performance of different models with stronger network backbone (ResNet) on the SmallNORB dataset. (b)(c) Confusion matrices of the predictions made by CNN and CNN$^2$ on the RGBD-Object dataset.

neither of the ResNet variants generalizes better than CNN$^2$. A backbone like ResNet that is strong to make predictions at seen angles does *not* imply that it is strong at unseen angles, and it can still benefit from CNN$^2$ to have improved 3D viewpoint generalizability. **Confusion Matrices.** Finally, we investigate how the predictions made by CNN$^2$ differ from those of vanilla CNN. Figures 9(b)(c) show the confusion matrices of the predictions made by CNN and CNN$^2$ at unseen view angles on the RGBD-Object dataset. The CNN$^2$ outperforms CNN in most cases, except when classifying the classes 1 (flashlight) and 4 (stapler) that are similar in shape but different in texture at certain view angles. This suggests that the CNN$^2$ relies more on shapes than textures to generalize, a bias that humans have been shown to possess (Geirhos et al. (2019)).

## 5 Conclusion

We propose the CNN$^2$ that gives improved 3D viewpoint generalizability of CNNs via a binocular vision. The CNN$^2$ uses dual feedforward pathways, recursive parallax augmentation, and the concentric multi-scale pooling to learn stereoscopic features. One important research direction following our work is to understand and visualize what have been learned by the filters and how they relate to that of human visualization. Furthermore, it would be interesting to apply CNN$^2$ to applications wherein a generalized vision system is highly in-demand, such as in self-driving cars.

# 6 Acknowledgments

This work is supported by the MOST Joint Research Center for AI Technology and All Vista Healthcare, Taiwan (MOST 108-2634-F-007-003-). We also thank the anonymous reviewers for their insightful feedbacks.

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
