[Supplementary Material]

# CNN$^2$: Supplementary Materials

**Wei-Da Chen**
Department of Computer Science
National Tsing-Hua University
Taiwan, R.O.C.
wdchen@datalab.cs.nthu.edu.tw

**Shan-Hung Wu**
Department of Computer Science
National Tsing-Hua University
Taiwan, R.O.C.
shwu@cs.nthu.edu.tw

## 1    Related Works

Here, we review some related work on improving the transformation generalizability of CNNs and the work on binocular vision.

**2D Transformations.** The vanilla CNNs generalizes poorly to transformed images beyond translation. Many studies (Jaderberg et al. (2015); Maninis et al. (2016); Cheng et al. (2016); Laptev et al. (2016); Worrall et al. (2017); Hinton et al. (2018)) have proposed techniques to improve the performance of CNNs when the test images have undergone 2D transformations like rotation, scaling, and sheering. The work (Jaderberg et al. (2015); Jia et al. (2016); Henriques and Vedaldi (2017)) explicitly allows these transformations of data within the network such that the filters can extract spatially invariant features. The studies (Cohen and Welling (2016); Zhou et al. (2017); Worrall et al. (2017); Cheng et al. (2019)) enhance regular convolution layers such that each activation map becomes equivariant to translations as well as rotations and reflections. The studies (Cohen and Welling (2017); Weiler et al. (2018); Ecker et al. (2019)) use steerable filters to learn rotation-equivariant CNNs.

**3D Viewpoint Generalization.** Getting the 3D viewpoint generalizability from 2D images is a challenging goal for model designers. Hinton et al. (2011) first propose the idea of capsule and use a non-convolutional network to learn group-invariant features. Later on, the study (Sabour et al. (2017)) enhances the idea of capsules by introducing a new model architecture, called capsule networks, which encodes an entity in a capsule that outputs a vector rather than a scalar. Different capsules are organized in a parse tree where the output of lower-level capsules is dynamically routed to upper-level capsules using a routing-by-agreement algorithm. The study (Hinton et al. (2018)) proposes the notion of matrix capsule that captures the activation (similar to that of a neuron) along with a pose matrix. A matrix capsule can learn to represent the relationship between the entity and the viewer (the pose). It then uses an EM-based routing algorithm to train the capsule network. As compared to CNN, the model achieves remarkable performance in 3D viewpoint generalization in the SmallNORB benchmark. However, capsule networks generally takes longer time to train due to a larger number of weights and the iterative routing algorithm. Furthermore, they are *not* compatible with existing CNN-based specialization techniques for various applications.

Another way to viewpoint generalization is by voxel discretization or by reconstructing manifold (and non-manifold) surfaces in 3D space from point clouds using voxels as an intermediate representation. However, models (Su et al. (2015); Qi et al. (2016, 2017)) that output voxels require either the voxel-level supervision or omnidirectional images as input, which are both expensive to get in practice. Yan et al. (2016) propose the perspective transformer nets that learn single-view 3D object reconstruction without 3D (voxel-level) supervision. Although being able to reconstruct 3D object from non-omnidirectional input images when there is only one class of objects, the model still requires omnidirectional images for leaning the reconstruction of multiple objects in different classes.

**Binocular Vision.** Prior to this paper, LeCun et al. (2004) have proposed a CNN model that takes two images as input. Figure 1 in the main paper shows its architecture, where the two images are stacked up along the channel dimension and then scanned by filters just like in normal CNNs. On the contrary,

the CNN$^2$ proposed in this paper introduces and explicitly models some priors from binocular vision that will be shown to significantly improve the 3D viewpoint generalizability of the model in our experiments. Binocular images have also been used for learning the depth information. The study (Godard et al. (2017)) utilizes the binocular images to learn the depth map in an unsupervised manner. The work (Kendall et al. (2017)) exploits the geometry and context information in binocular images to learn the disparity map of a stereogram.

**Multi-Scale Feature Representations.** A CNN$^2$ layer extracts features at multiple scales, thus is related to the work on multi-scale feature learning (Yang and Ramanan (2015); Cai et al. (2016); Lin et al. (2017); Chen et al. (2019)). DAG-CNN (Yang and Ramanan (2015)), MSCNN (Cai et al. (2016)) and FPN (Lin et al. (2017)) use features learned at different CNN layers to form multi-scale features. Recently, Chen et al. (2019) use multiple CNN branches to form multi-resolution pathways. Unlike the above models that simply concatenate the multi-scale features to make predictions, the CNN$^2$ pools (via the CM pooling) multi-scale features to make them of equal size and then stacks them up along the channel dimension. The location information encoded in different feature maps are aligned. This allows the next convolution layer to learn location independent patterns (and a sense of 3D dimensionality) by contracting the features at different scales.

**Pooling Strategy.** Pooling layers, such as the max pooling layers, are commonly used by a CNN to reduce the size of presentations along the spatial dimensions and to speed up computation. However, a pooling layer may introduce translation invariance that decreases viewpoint generalizability (Hinton et al. (2011); Sabour et al. (2017)). A CM pooling layer proposed in the main paper does *not* introduce translation invariance because its stride is always 1 at all scales. Our CM pooling is cosmetically similar to some existing pooling techniques (He et al. (2014); Gong et al. (2014); Qi et al. (2018)). The spatial pyramid pooling (He et al. (2014)) pools image pixels using predefined patches, which require domain-specific knowledge to define. The multi-scale orderless pooling (Gong et al. (2014)) outputs feature maps of different sizes, but these maps are not "zoomed" to equal size and then stacked up along the channel dimension to help the filters contract features at different scales at the same location. Qi et al. (2018) propose a concentric circle pooling strategy to achieve rotation invariance, where multiple filters scan an image/feature map following concentric window-sliding paths. Their term "concentric" is different from ours.

## 2   Correspondence to Human Visual System

Figure 1 shows an oversimplified version of the mammals' visual system. The visual information mainly flows through the *central visual pathways* in the brain.[1] The retina in each eye first transduces the light reflected by the cornea and lens into electrical pulses. The optic nerve then carries these pulses and sends them to the *optic chiasma* where the information coming from both eyes is combined and then splits. Then the two halves of the split information are sent to the left and right halves of the brain, respectively. At each side, the information flows through the lateral geniculate nucleus (LGN), a sensory relay nucleus, and finally reaches the *visual cortex system* where the visual image is heavily processed. Within the system, the visual information flows through a (ventral) cortical hierarchy consisting of V1, V2, V4, and the inferior temporal (IT) cortex areas that extract simple to complex object features, respectively. Recent studies (Kheradpisheh et al. (2016); Wallis et al. (2017); Laskar et al. (2018); Long and Konkle (2018)) have found correspondence between the activations of CNN layers and the neuron responses in these areas.

However, the CNNs are different from human visual system in many ways. One key difference is that the CNNs have only one feedforward pathway. On the other hand, the CNN$^2$ employs two feedforward pathways, which resembles the left and right halves of the central visual pathway in two sides of our brains. The dual parallax augmentation at the input layer of the CNN$^2$ corresponds to the optic chiasma in human visual system, where the information coming from both eyes is combined, augmented, and then splits. At deep layers, it resemble the interactions between the left and right sides of the brain which are known to have their own biases (Gotts et al. (2013)).

The concentric multi-scale pooling layers in CNN$^2$ are also inspired by the brain. Although serving much the same function as V1, the V2 cortex area is found to be able to handle depth extraction (Von

Figure 1: Visual system of mammals. The electrical pulses from the two eyes are merged at optic chiasma and then sent to the right and left brains separately following two visual pathways and finally reach the visual cortex system where the visual image is heavily processed by the interaction between the right and left brains with respective bias.

(a) All classes  (b) 5 classes

Figure 2: 3D viewpoint generalizability of models at different view angles on the RGB-D dataset. Only the first one third view angles are seen at training time. (a) Test accuracy of models is misled by objects in different classes that look similar at some particular view angles. (b) Test accuracy of models on 5 selected classes: "camera," "flashlight," "lightbulb," "pitcher," and "stapler."

Der Heydt et al. (2000)), foreground distinguishment (Qiu and Von Der Heydt (2005); Maruko et al. (2008)), and illusory contours (von der Heydt et al. (1984); Anzai et al. (2007)). These are exactly what we expect a filter in $CNN^2$ to detect (in addition to 2D object patterns) when scanning through the output $\dot{h}$ of an CM pooling layer.

While the design of $CNN^2$ model is loosely inspired by how the vision system in humans' brains work, the effectiveness of $CNN^2$ solely depends on engineering efforts.

## 3 More on Experiments

### 3.1 Datasets and Preprocessing

We conduct experiments using the following datasets:

**SmallNORB** (LeCun et al. (2004)). This dataset consists of 48,600 grayscale binocular images of size $96 \times 96$ pixels in 5 classes: four-legged animals, human figures, airplanes, trucks, and cars. Each class has 10 object instances, and each instance has 972 binocular images taken under 6 lighting conditions, 9 elevations (30 to 70 degrees, with ticks at each 5 degrees), and 18 azimuths (0 to 340, with ticks at every 20 degrees).

**ModelNet2D**. The ModelNet40 dataset (Wu et al. (2015)) contains about 12,311 CAD models in 40 classes. We randomly pick 5 classes: chairs, human figures, cars, airplanes, and lamps. Each class has 15 model instances. Following the settings used in the study (LeCun et al. (2004)), we render the each model into 648 grayscale binocular images with 72 azimuth angles (with ticks at each 5 degrees), 9 elevations (30 to 70 degrees with 5-degree ticks) under the fixed lighting conditions. The distance between the two lens in the binocular camera is set to 7.5 cm, and the resolution of each images is $96 \times 96$ pixels. Finally, we obtain 48,600 binocular images. We call this image set the ModelNet2D.

**RGB-D Object** (Lai et al. (2011)). This dataset includes 300 common household objects in 51 classes. It was recorded using a Kinect style 3D camera that records synchronized and aligned 640x480 RGB and depth images. Each object was placed on a turntable and video sequences were captured for one whole rotation. For each object, there are 3 video sequences, each recorded with the camera mounted at a different height so that the object is viewed from different angles with the horizon. We take images of the same object with angles at multiples of 10 degrees and get 250,000 colored images in total.

Unlike the SmallNORB dataset where the 5 classes were carefully selected such that objects in different classes look different at most view angles, the ModelNet2D and RGB-D Object datasets contain objects in different classes that are hard to distinguish by humans at many view angles, which leads to degraded performance, as shown in Figure 2(a). To prevent this from misleading our study on 3D viewpoint generalizability of different models, we select 5 classes from the ModelNet2D and RGB-D Object datasets such that objects in different classes can be distinguished by humans at every view angle. Specifically, we select the "chair," "human," "airplane," "car," and "lamp" classes from the ModelNet2D dataset and the "camera," "flashlight," "lightbulb," "pitcher," and "stapler" classes from the RGB-D Object dataset. Focusing on these classes only, we get performance of different models that is aligned with the task difficulty at different view angles, as shown in Figure 2(b).

## 3.2 Training and Model Architectures

To speed up the training process, we downsample SmallNORB and ModelNet2D images to $48 \times 48$ pixels and RGB-D Object images to $72 \times 72$ pixels and $z$-normalize each pixel. We train each model using Adam algorithm (Kingma and Ba (2014)) with mini-batches of size 32. We use early stopping strategy for all experiments.

For the grayscale datasets (ModeNet2D and SmallNORB), we try to adhere to the architectures and hyperparameters used by the original papers when training different models in order to allow fair comparison. Following the settings in LeCun et al. (2004), the vanilla CNN has three convolutional layers with 16, 32 and 64 filters respectively. Each filter has size $5 \times 5$ with a stride 1 and is followed by a $2 \times 2$ max pooling with stride 2. The task model is a fully connected layer with 1024 units and dropout and is connected to the 5-way softmax output layer. All units use the ReLU as the activation function. The CapsuleNet and PTN follow the architecture and settings described in the original papers (Hinton et al. (2018); Yan et al. (2016)). The Monodepth uses the model described in the original paper (Godard et al. (2017)) to output the depth map in stage 1. Then, in stage 2, we add the depth map into the the left eye image as an additional channel and feed the augmented image to a CNN. The CNN architecture is the same as that used in the Vanilla CNN. In the BL-Net, a BL-module includes $K$ branches where the $k^{th}$ branch represents an image of scale $1/2^k$. We set $K$ (number of branches) to 2, $\alpha$ (reduction in number of channels) to 2, and $\beta$ (reduction in number of layers) to 4. Our CNN[2] consists of 3 blocks. Each block has a CM pooling layer and a convolution layer. We use 3 scales ($s = 0, 1, 2$) in a CM pooling layer. The first, second, and third convolution layers have 6, 12, and 32 filters with sizes $5 \times 5$, $5 \times 5$, and $3 \times 3$ respectively. All filters use the stride of 1, and each unit uses the ReLU non-linearity.

For the larger-scale RGB-D Object dataset, there is no known "good settings" for each of the considered models. We use grid search to obtain a good combination of hyperparameters for each model. Unfortunately, we did not find "good" configurations for CapsuleNet and PTN that lead to satisfactory 3D viewpoint generalizability. We detail our search below:

**Vanilla CNN.** We use deeper layers in the candidates to fit the RGB-D Object dataset, since it has larger feature space than grayscale datasets. Table 1 summarizes our model candidates of vanilla CNN. The vanilla CNN gives relatively stable performance during the hyperparameter search.

Table 1: Candidate architectures of vanilla CNN. We use convolutional layer as task model. It shows number of filters, followed by the filter size.

| # Channels | | | | | Task Model | | Training Acc. |
|---|---|---|---|---|---|---|---|
| 16(5) | 16(5) | 32(3) | 32(3) | 32(3) | 256(2) | 256(1) | 0.776 |
| 16(5) | 16(5) | 32(3) | 32(3) | 32(3) | 512(2) | 512(1) | 0.781 |
| 16(5) | 16(5) | 32(3) | 48(3) | 48(3) | 512(2) | 512(1) | 0.788 |
| **16(5)** | **16(5)** | **48(3)** | **48(3)** | **48(3)** | **512(2)** | **512(1)** | **0.795** |
| 16(5) | 16(5) | 32(3) | 48(3) | 64(3) | 512(2) | 512(1) | 0.783 |

Table 2: Candidate architectures of CapsuleNet. $A$ ($s$): channels of first convolutional layer with stride $s$. $B$: number of primary capsule types. $P$: pose size. $C_i(s)$: number of capsule types in the $i$-th convolutional capsule layers with stride $s$. $K_i$: size of the receptive field in the $i$-th (capsule) convolutional layer.

| $A$ ($s$) | $B$ | $P$ | $C$ ($s$) | | | $K$ | | | | Training Acc. |
|---|---|---|---|---|---|---|---|---|---|---|
| 32 (5) | 32 | 4 | 32 (2) | 32 (2) | | 5 | 3 | 3 | | 0.337 |
| 64 (5) | 32 | 4 | 32 (2) | 32 (2) | | 5 | 3 | 3 | | 0.401 |
| 64 (5) | 32 | 4 | 32 (2) | 32 (2) | | 5 | 5 | 3 | | 0.415 |
| 64 (5) | 32 | 4 | 32 (2) | 32 (2) | | 5 | 5 | 5 | | 0.517 |
| 64 (5) | 32 | 4 | 64 (2) | 32 (2) | | 5 | 5 | 5 | | 0.408 |
| 64 (5) | 32 | 4 | 64 (2) | 64 (2) | | 5 | 5 | 5 | | 0.392 |
| 64 (5) | 32 | 4 | 32 (2) | 32 (2) | 32 (2) | 5 | 5 | 5 | 5 | 0.411 |
| 64 (5) | 48 | 4 | 32 (2) | 32 (2) | 32 (2) | 5 | 5 | 5 | 5 | 0.508 |
| 64 (5) | 48 | 4 | 32 (2) | 32 (2) | 32 (2) | 3 | 3 | 5 | 5 | 0.537 |

**CapsuleNet.** The model starts with a convolutional layer with $A$ filters, $K$ receptive filed, and a stride of $s$ with the ReLU activation followed by the the primary capsule layer. The $P \times P$ pose of each of the $B$ primary capsule types encode a learned linear transformation of lower-layer activations. The activations of the primary capsules are produced by applying the sigmoid function to the weighted sums of the same set of lower-layer ReLUs. The primary capsules are followed by $i$ convolutional capsule layers (with size $K \times K$ in spatial dimensions), each with $C_i$ capsule types and a stride of $s$. The last layer of the convolutional capsules is connected to the final capsule layer which has one capsule for each output class. Table 2 shows the training accuracy of the CapsuleNet for the experiments described in Section 4.1 of the main paper with different configurations. We as can see, the CapsuleNet gives poor training accuracy in all cases. Peer et al. (2018) have pointed out that the capsule networks are hard to train on complex datasets and the iterative routing-by-agreement algorithm used for training does not ensure the emergence of a parse tree in the networks. Our results seem to validate this.

**PTN.** We vary the architecture described by Yan et al. (2016) to get candidates. Table 3 shows the training accuracy of all model architectures we have considered. Without being trained on omnidirectional images, the PTN cannot reconstruct the voxel of objects and then use them to make

Table 3: Candidate architectures of PTN. In 2D and 3D convolution layers, $C(K, s)$ denotes $C$ filters with the receptive field of size $K \times K$ and stride of $s$.

| Encoder | | | | | | Decoder | | | | | Training Acc. |
|---|---|---|---|---|---|---|---|---|---|---|---|
| 2D convolution | | | Fully-connected | | | Fully-connected | | 3D convolution | | | |
| 64(5,2) | 128(5,2) | 256(5,2) | 1K | 1K | 512 | | 512×3×3×3 | 256(4,2) | 96(5,2) | 1(6,2) | 0.401 |
| 128(5,2) | 128(5,2) | 256(5,2) | 1K | 1K | 512 | | 512×3×3×3 | 256(4,2) | 96(5,2) | 1(6,2) | 0.415 |
| 128(5,2) | 128(5,2) | 256(5,2) | 1K | 1K | 1K | | 512×3×3×3 | 256(4,2) | 96(5,2) | 1(6,2) | 0.517 |
| 128(5,2) | 128(5,2) | 256(5,2) | 1K | 1K | 1K | 1K | 512×3×3×3 | 256(4,2) | 96(5,2) | 1(6,2) | 0.408 |
| 128(5,2) | 128(5,2) | 256(5,2) | 1K | 1K | 1K | 1K | 512×3×3×3 | 256(4,2) | 128(5,2) | 1(6,2) | 0.392 |
| 128(5,2) | 128(5,2) | 256(5,2) | 1K | 1K | 1K | 1K | 512×3×3×3 | 256(4,2) | 256(5,2) | 1(6,2) | 0.457 |
| 128(5,2) | 256(5,2) | 256(5,2) | 1K | 1K | 1K | 1K | 512×3×3×3 | 256(4,2) | 256(5,2) | 1(6,2) | 0.521 |

Table 4: Candidate architectures of CNN$^2$. The $i$-th $s$ denotes the $i$-th scale level in a concentric multi-scale pooling layer, and the $i$-th #Channels denotes the number of filters used in the $i$-th layer.

| $s$ | | | #Channels | | | | Test Acc. at $0°$ | $90°$ | $180°$ | $270°$ | Unseen Avg. |
|---|---|---|---|---|---|---|---|---|---|---|---|
| 1 | 3 | 5 | 6 | 12 | 32 | | 1.0 | 0.671 | 0.648 | 0.623 | 0.631 |
| 1 | 3 | 5 | 12 | 12 | 32 | | 1.0 | 0.690 | 0.751 | 0.717 | 0.836 |
| 1 | 3 | 5 | 12 | 24 | 32 | | 1.0 | 0.715 | 0.772 | 0.725 | 0.761 |
| 1 | 3 | 5 | 12 | 24 | 64 | | 1.0 | 0.712 | 0.769 | 0.721 | 0.753 |
| 1 | 3 | 5 | 12 | 24 | 48 | | 1.0 | 0.721 | 0.776 | 0.734 | 0.772 |
| 0 | 1 | 2 | 12 | 24 | 32 | | 1.0 | 0.811 | 0.814 | 0.809 | 0.810 |
| 0 | 1 | 2 | 12 | 24 | 64 | | 1.0 | 0.797 | 0.801 | 0.784 | 0.801 |
| 0 | 1 | 2 | 12 | 24 | 48 | | 1.0 | 0.823 | 0.817 | 0.820 | 0.824 |
| 0 | 1 | 2 | 12 | 24 | 24 | 24 | 1.0 | 0.780 | 0.794 | 0.817 | 0.811 |
| 0 | 1 | 2 | 12 | 24 | 24 | 40 | 1.0 | 0.699 | 0.768 | 0.822 | 0.767 |
| 0 | 1 | 2 | 12 | 24 | 40 | | 1.0 | **0.861** | **0.831** | **0.852** | **0.868** |

(a) SmallNORB

(b) SmallNORB

(c) RGB-D Object

(d) RGB-D Object

Figure 3: 3D viewpoint + 2D rotation generalizability of models trained on the SmallNORB and RGB-D Object datasets. (a)(c) Test accuracy of different models for objects with unseen rotations. Legends follow Figure 2. (b)(d) CNN$^2$ is backward compatible with existing CNN-based techniques and can be readily enhanced to have 2D rotation generalizability.

predictions. We suspect that the sample complexity of PTN is too high for the binocular images in RGB-D Object dataset to guide the training properly.

**CNN$^2$.** While training on the larger-scale RGB-D Object dataset seems too challenging for CapsuleNet and PTN, the CNN$^2$ can learn to generalize from it without much difficulty. Here, we search for an CNN$^2$ architecture larger than the one trained on the SmallNORB and ModelNet2D datasets in order to accommodate the nuance of channel patterns in colored images. Table 4 shows the model candidates and their respective performance on 3D viewpoint generalization. As we can see, every candidate makes perfect predictions at seen view angles, and most candidates achieve satisfactory test accuracy at unseen view angles. We use the architecture shown in the last row to compare it with other baselines in the main paper.

Figure 4: Performance of different models on the ShapeNet dataset where each class contains at least 100 object instances.

## 3.3 Backward Compatibility

In section 4.2 of the main paper, we claimed that the $CNN^2$, which was designed for 3D viewpoint generalizability, can incorporate existing CNN-based techniques (Jaderberg et al. (2015); Cohen and Welling (2016); Worrall et al. (2017)) to have enhanced 2D rotation generalizability. Following the same experiment settings where different models are used to predict the labels of test images with unseen viewpoint translation *and rotation* (see section 4.2 of the main paper for more details), we demonstrate that the above claim holds on the SmallNORB and RGB-D Object datasets as well. As shown in Figure 3, the "enhanced" $CNN^2$ give significantly better performance at all view angles. The $CNN^2$ performs comparably with CapsuleNet on the SmallNORB dataset, and much better on the RGB-D Object dataset.

The above example demonstrates how the $CNN^2$ could benefit existing work. By integrating with other techniques in the rich CNN ecosystem, the $CNN^2$ has great potential to impact on many further applications.

## 3.4 Scaling Up the Number of Instances

In this section, we scale up the number of instances in each class by conducting new experiments using the ShapeNet dataset Chang et al. (2015) following the settings for ModelNet2D. ShapeNet is a richly-annotated and large-scale dataset of 3D shapes, covering 55 common object categories with about 51,300 unique 3D models. We randomly sample 5 classes (airplanes, cars, cameras, lamps, and chairs) as our target predictions. Now, each class has at least 100 instances, and Figure 4 shows the results. The increased number of instances in each class does not seem to make CapsuleNet easier to train, as we still did not get satisfactory performance. On the other hand, $CNN^2$ can readily scale up to a larger dataset and consistently improve the generalizability of vanilla CNN.

## Footnotes

[1]In fact, the vision system has multiple (forward and backward, dorsal and ventral) visual pathways and is much more complex than the one presented here. Interested readers may refer to an in-depth summary (Milner and Goodale (2006)) for more details.