[Reviews · NeurIPS 2019]

Reviewer 1



The paper is well written. The explanation is easy to follow. The proposed approach is well-motivated. Ln. 129: the paper states that CNN^2 need to use far fewer filters to achiever performance comparable to existing schemes. Does the accuracy of CNN^2 improve as more filters are added. Am I correct to assume that the size of the features remain the same as the input as one moves up the layers due to concentric multi-scale pooling? Is it necessary?

Reviewer 2



Originality: To my knowledge, the motivation for such dual-pathway design is not new. But the particular design of this paper, CM polling in particular, is definitely novel. Quality: I think the evaluation of this work is quite thorough, but missing some important items. 1. Missing an important ablation study. It seems that using CM pooling in vanilla CNNs is not not shown in the paper. This makes it less clear if the this pooling actually improves the performance of vanilla CNNs. 2. Missing Vanilla CNN tuning details. It is great that the authors provided the hyper-parameter search details for PTN, CNN2 and Capsule nets. Yet it seems unclear how the vanilla CNNs are tuned. It seems that hyper-parameter tuning plays a significant role in CNN2's performance(6.7% performance drop by changing the last channel# from 40 to 64). 3. Scales of the study. The number of instances in each class is so small ~15-20, which might lead to great variance. This makes it hard to interpret the numbers. 4. Missing confusion matrices. Having a confusion matrices would make the numbers more interpretable as it provides details on the error patterns between classes. Clarity: This paper is well written with significant and thorough reference to related works. Significance: Personally, I believe this direction of incorporating biologically inspired inductive bias into network designs, and the particular problem of generalizing object recognition across vastly different views are of great significance. However flaws in the evaluation sections do have a negative impact on this.

Reviewer 3



With the above contributions, the paper explores the problem of paired-image classification. It designs plausible ways for the two streams of the network to interact so as to simulate the binocular vision. However, the connection between the network architecture and human binocular vision system is far-fetched. The simple minus and concatenation do not seem to be a generic way to pass messages across streams of networks. In particular, the effectiveness of the proposed modules is not well-justified. It is not surprising at all that a two-stream network with across-stream interaction can perform better than an early fusion network. How about the late fusion networks and other fusion methods that pass messages between the two streams? Without comparison more possible baselines, it is not possible to experimentally justify the proposed modules are particularly useful for binocular images. The experimental settings are toy-like, and the neural network backbone is also weak. Results under these settings are not practically meaningful. When a strong neural network can get much better accuracy using a single image in practice, it is not so useful to use a suboptimal binocular system to get some improvement regarding weak baselines on non-realistic datasets.

[Author Response · NeurIPS 2019]

**To Reviewer 1.** Thanks for your positive comments. **Q1:** How the proposed architecture would fair on ... single-image object classification? **A1:** Good question! With monocular images, the parallax channels contain all zeros, therefore the $CNN^2$ degenerates into a conventional CNN gracefully. Fig. (a) shows the performance of degenerated $CNN^2$ with the single-eye images from the RGB-D Object dataset. **Q2:** Does the accuracy of $CNN^2$ improve as more filters are added? **A2:** As shown in Table 3 in the supplementary file, increasing the number of filters in $CNN^2$ does not guarantee performance gain. **Q3:** Does the size of the features remain the same as the input as one moves up the layers due to CM pooling? Is it necessary? **A3:** The size does not need to be the same as the input nor across layers, but the size of feature maps for the same filter must be of the same size in order to allow the filter to detect stereoscopic patterns.

(a)

**To Reviewer 2.** Thanks for your constructive comments.
**Q1:** Ablation study of CM pooling on vanilla CNN. **A1:** Fig. (b) shows the performance of vanilla CNN with CM pooling over the RGB-D Object dataset. The CM pooling can indeed help the vanilla CNN detect useful features. **Q2:** Vanilla CNN tuning details. **A2:** The table below summarizes our model candidates for the RGB-D Object dataset. The vanilla CNN gives relatively stable performance during the hyperparameter search. **Q3:** Scale up the number of instances in each class ... using ShapeNet. **A3:** As suggested, we conduct new experiments using the ShapeNet dataset following the settings for ModelNet2D. Now, each class (airplanes, cars, cameras, lamps, and chairs) has at least 100 instances. Fig. (c) shows that $CNN^2$ still outperforms CNN and CapsuleNet.
**Q4:** Confusion matrices for classification. **A4:** Below please see the confusion matrices of the predictions made by CNN and $CNN^2$ at unseen view angles on the RGBD-Object dataset. The $CNN^2$ outperforms CNN in most cases, except when classifying the classes 1 (flashlight) and 4 (stapler) that are similar in shape but different in texture at certain view angles. This suggests that the $CNN^2$ relies more on shapes than textures to generalize, a bias that humans have been shown to possess (Matthias Bethge et al., "ImageNet-trained CNNs are biased towards texture; increasing shape bias improves accuracy and robustness," ICLR'19).

(b)

(c)

| | # Channels | | | | Task Model | | Unseen Avg. |
|---|---|---|---|---|---|---|---|
| 16(5) | 16(5) | 32(3) | 32(3) | 32(3) | 256(2) | 256(1) | 0.776 |
| 16(5) | 16(5) | 32(3) | 32(3) | 32(3) | 512(2) | 512(1) | 0.781 |
| 16(5) | 16(5) | 32(3) | 48(3) | 48(3) | 512(2) | 512(1) | 0.788 |
| **16(5)** | **16(5)** | **48(3)** | **48(3)** | **48(3)** | **512(2)** | **512(1)** | **0.795** |
| 16(5) | 16(5) | 32(3) | 48(3) | 64(3) | 512(2) | 512(1) | 0.783 |

**To Reviewer 3.** Thanks for your comments. **Q1:** How about the late fusion networks...? **A1:** As suggested, we compare $CNN^2$ with two new baselines that perform early and late "fusion" (i.e., dual parallax augmentation) in only the first and last layer, respectively. Fig. (d) shows the results on the RGB-D Object dataset. $CNN^2$ outperforms other baselines because it has fusion at all layers, which allows small differences between the feature maps in two paths to add up to a big difference at a deeper layer. **Q2:** The experimental settings are toy-like. **A2:** Please note that our classification tasks are tested at view angles that are *unseen* during the training time. Comparing to traditional image classification, these tasks are very challenging, and our settings are already more complex than the ones used by Hinton et al. in their CapsuleNet work published in ICLR'18, which considered only grayscale images. The $CNN^2$ has advanced the state-of-the-art performance on the grayscale ModelNet2D and SmallNORB datasets and, for the first time, gives improved 3D viewpoint generalizability on the colored RGB-D Object dataset. **Q3:** The neural network backbone is weak. **A3:** A backbone, e.g. ResNet, that is strong to make predictions at seen angles does *not* imply that it is strong at unseen angles. To show this, we compare the performance of $CNN^2$ with ResNet-50 and a toy ResNet having a similar number of parameters as $CNN^2$ on the SmallNORB dataset. The results are shown in Fig. (e). Although the SmallNORB dataset contains only grayscale images and looks "easy," neither of the ResNet variants generalizes better than $CNN^2$. We will add the above experiments to the paper.

(d)   (e)

We hope our above explanation relieves your concerns, and if so, please consider raising your score.

[Meta-Review · NeurIPS 2019]

This paper addresses a challenging problem: classification on unseen viewpoints by training on seen viewpoints. The authors propose a novel model inspired by human binocular vision to addess this problem. Generally speaking, the paper is well organized and motivated. The evaluation part is a bit weak, but the rebuttal provided by the authors has remedy this issue. This work would be an interesting work for the computer vision community. The authors are encouraged to address the issues mentioned by the reviewers.